# Modulation of Antigen Display on PapMV Nanoparticles Influences Its Immunogenicity

**DOI:** 10.3390/vaccines9010033

**Published:** 2021-01-08

**Authors:** Marie-Eve Laliberté-Gagné, Marilène Bolduc, Caroline Garneau, Santa-Mariela Olivera-Ugarte, Pierre Savard, Denis Leclerc

**Affiliations:** 1Department of Microbiology, Infectiology and Immunology, Faculty of Medicine, Laval University, Quebec City, QC G1V 4G2, Canada; Marie-Eve.L-Gagne@crchudequebec.ulaval.ca (M.-E.L.-G.); Marilene.Bolduc@crchudequebec.ulaval.ca (M.B.); Caroline.Garneau@crchudequebec.ulaval.ca (C.G.); Santa-Mariela-Olivera.Ugarte@crchudequebec.ulaval.ca (S.-M.O.-U.); 2Department of Molecular biology, medical biochemistry and pathology, Faculty of Medicine, Laval University, Quebec City, QC G1V 4G2, Canada; Pierre.Savard@crchudequebec.ulaval.ca

**Keywords:** vaccine platform, papaya mosaic virus (PapMV), rod-shaped nanoparticle, influenza M2e, influenza nucleocapsid, sortase (SrtA)

## Abstract

Background: The papaya mosaic virus (PapMV) vaccine platform is a rod-shaped nanoparticle made of the recombinant PapMV coat protein (CP) self-assembled around a noncoding single-stranded RNA (ssRNA) template. The PapMV nanoparticle induces innate immunity through stimulation of the Toll-like receptors (TLR) 7 and 8. The display of the vaccine antigen at the surface of the nanoparticle, associated with the co-stimulation signal via TLR7/8, ensures a strong stimulation of the immune response, which is ideal for the development of candidate vaccines. In this study, we assess the impact of where the peptide antigen is fused, whether at the surface or at the extremities of the nanoparticles, on the immune response directed to that antigen. Methods: Two different peptides from influenza A virus were used as model antigens. The conserved M2e peptide, derived from the matrix protein 2 was chosen as the B-cell epitope, and a peptide derived from the nucleocapsid was chosen as the cytotoxic T lymphocytes (CTL) epitope. These peptides were coupled at two different positions on the PapMV CP, the N- (PapMV-N) or the C-terminus (PapMV-C), using the transpeptidase activity of Sortase A (SrtA). The immune responses, both humoral and CD8+ T-cell-mediated, directed to the peptide antigens in the two different fusion contexts were analyzed and compared. The impact of coupling density at the surface of the nanoparticle was also investigated. Conclusions: The results demonstrate that coupling of the peptide antigens at the N-terminus (PapMV-N) of the PapMV CP led to an enhanced immune response to the coupled peptide antigens as compared to coupling to the C-terminus. The difference between the two vaccine platforms is linked to the enhanced capacity of the PapMV-N vaccine platform to stimulate TLR7/8. We also demonstrated that the strength of the immune response increases with the density of coupling at the surface of the nanoparticles.

## 1. Introduction

Vaccines are recognized as the greatest breakthrough in medical science, and they have saved more human lives than any other medical approach [1,2,3,4,5,6]. Most current successful vaccines are made using conventional techniques based on either inactivated or attenuated pathogens, which do not adapt well to a pandemic situation [7]. To face new viral threats, there is a medical need for vaccine platforms that can be deployed rapidly to face such challenges [8]. Recently, nucleic-acid-based vaccine platforms have proven very successful in the development of COVID-19 vaccines [9,10,11,12,13]. Nevertheless, other vaccine technologies are still needed to lower the cost of production and ensure distribution without the need for a cold chain. Importantly, vaccine platforms that can elicit a balanced humoral and CD8+ T-cell-mediated immune response, capable of providing broad protection to conserved antigens, are still today a priority and a major need [7,13,14,15,16].

Lately, vaccine platforms based on the use of a rod-shaped nanoparticle made from the coat protein (CP) of plant viruses has triggered much interest [17,18,19,20,21,22,23,24]. One of these vaccine platforms, the papaya mosaic virus (PapMV) nanoparticle, emerge as an very interesting technology because it was showed to be a potent Toll-like receptor 7 and 8 (TLR 7/8) agonist [25] that triggers innate immunity [26] leading to secretion of type 1 interferon (IFN-) [25,26,27,28] and the triggering of the antiviral cellular arsenal. The genetic fusion of small peptides directly to the open reading frame of the PapMV CP leads to the generation of vaccine candidates capable of eliciting both a humoral [28,29] and a CD8+-mediated immune response [30] through induction of the cross-presentation of the CTL epitope on major histocompatibility complex (MHC) class 1 [31]. This approach is interesting but is also limiting because fusion of the peptide can interfere with the structure of the CP, leading to a chimeric CP that is unable to self-assemble into nanoparticles, the formation of which is critical to obtain a good immune response.

To circumvent this problem, a novel approach, in which the peptide is fused directly onto self-assembled nanoparticles [32,33] using a bacterial transpeptidase—sortase A (SrtA)—was developed [34,35]. This approach is more flexible than genetic fusion to the CP, and it allows the fusion of larger peptides (>39 amino acids) [33] and even full-length proteins [35]. In each case, attachment of the antigens to the PapMV vaccine platform significantly improved the humoral and/or the CD8+ -mediated immune response to the coupled peptide antigen [33,35]. The SrtA enzyme joins the receptor motif (LPETGG), genetically fused to the PapMV CP (PapMV-C), to the donor motif provided by the antigen (repeated G motif) [33,35,36] generating a candidate vaccine. 

In this study, we engineered a new generation of PapMV vaccine platforms in which the coupling is made at the N-terminus (PapMV-N) instead of the C-terminus of the CP. We assessed the impact of the position of coupling of the antigen on the efficiency of both vaccine platforms (PapMV-C and the PapMV-N) in eliciting an immune response. In addition, we assessed the impact of the density of the antigen coupled to the nanoparticle on the immune response generated by different vaccine candidates.

## 2. Materials and Methods 

### 2.1. Sortase A and Nanoparticle Protein Production, Purification, and Self-Assembly

Methods for the production and purification of the Sortase A transpeptidase, PapMV, and PapMV-C have been reported previously [33]. The PapMV-N CP sequence, containing four glycines at its N-terminus and a 6His-tag at its C-terminus, was genetically inserted at the C-terminus of the intein sequence in the pTWIN1 vector (New England Biolabs Canada, Withby, Ontario, Canada) (Figure 1A). The fusion protein (49.3 kDa) was expressed in *Escherichia coli* (BL21) for 16 h at 16 °C and purified on an ion matrix affinity chromatography (IMAC) with Ni Sepharose 6FF resin (Cytiva, Canada, Vancouver, British Colombia). Intein self-cleavage was induced with a pH shift and a temperature shift: buffer exchange (from pH 8.0 to 6.5) followed by a 16–24 h incubation at room temperature. Self-cleavage releases two protein fragments of 24.1 kDa (PapMV-N) and 25.2 kDa (intein) that are revealed on 10% Tris–Tricine SDS-PAGE (Figure 1B). The intein fragment was removed on a second IMAC where only PapMV-N harboring the 6His-tag was retained on the beads. Endotoxins were removed from purified proteins with a strong quaternary ammonium membrane (Sartobind Q nano, Sartorius Stedim Biotech, Goettingen, Germany). 

All these various CP forms (PapMV, PapMV-C, and PapMV-N) were self-assembled with a 1517 nucleotide long ssRNA template, leading to the generation of rod-like nanoparticles as previously described [26]. 

### 2.2. Peptides 

The M2e peptide used for the coupling reaction on the PapMV-C nanoparticle harbors the GG donor motif at its N-terminus (GGSLLTEVETPIRNEWGCRCNDSSD). The M2e peptide used for the coupling reaction on the PapMV-N nanoparticle harbors the LPETGG receptor motif at its C-terminus (SLLTEVETPIRNEWGCRCNDSSDLPETGG). The peptides (purchased from GenScript (Piscataway, NJ, USA) and Biomatik (Cambridge, Ontario, Canada), respectively) were solubilized in sterile water or in 55 % DMSO. The M2e peptide used for the enzyme-linked immunosorbent assay (ELISA) (SLLTEVETPIRNEWGCRCNDSSD) was purchased from GenScript (Piscataway, NJ, USA) and solubilized in 0.1M PBS pH 7.4. 

The nucleocapsid protein (NP) peptides (GGGGGHSNLNDATYQRTRALVRTGMDPR for PapMV-C and HSNLNDATYQRTRALVRTGMDPRLPETGG for PapMV-N) were purchased from PepMic (Suzhou, Jiangsu, China). Both peptides were solubilized in sterile water. The NP peptide ^147^TYQRTRALV^155^ used for the interferon gamma enzyme-linked immune absorbent spot (ELISPOT) was purchased from Anaspec (Fremont, CA, USA) and solubilized in sterile water.

### 2.3. Biophysical Characterization of Platforms

PapMV, PapMV-C, and PapMV-N nanoparticles were analyzed by dynamic light scattering (DLS) to assess their mean length and their resistance to heat denaturation, as described previously [33]. DLS analyses were conducted using a ZetaSizer Nano ZS (Malvern Panalytical, Malvern, United Kingdom) and Zetasizer software (version 7.11) (Malvern Panalytical, Malvern, United Kingdom). Briefly, nanoparticle liquid suspensions were diluted at 0.2 mg/mL in formulation buffer; 100 µL was transferred to a microcuvette at 4 °C for assessment of the length of the nanoparticles (four measurements). The same procedures were used to assess the resistance of the nanoparticle to heat denaturation using temperatures ranging from 23 to 50 °C. The shape of the nanoparticles was analyzed by transmission electron microscopy (TEM) with a TECNAI Spirit G2 Biotwin 120 kV microscope (FEI, Hillsboro, OR, USA). Briefly, samples were diluted at 0.02 mg/mL in water prior to being stained for 7 min in 3% uranyl acetate solution; then, stained samples were transferred to a carbon formvar-coated grid and dried before TEM observation. 

### 2.4. Coupling Reactions with SrtA

SrtA conjugation reactions were performed in 1× SrtA reaction buffer (50 mM Tris–HCl pH 8.0, 150 mM NaCl, 10 mM CaCl_2_). In general, each reaction contained 25–100 μM of SrtA, 5–250 μM of the peptide to be coupled, and 25–100 μM of either the PapMV-C or the PapMV-N nanoparticle liquid suspension; reaction samples were then incubated for 2.5 h to 20 h at room temperature in a final volume of 1.7–13 mL. Reactions were stopped with the addition of EDTA to a final concentration of 10 μM. Prior to injection in animals, reactions were dialyzed in 10 mM Tris–HCl, 150mM NaCl pH 8.0 buffer on a 100 kDa molecular weight cutoff (MWCO) membrane (Float-A-Lyzer G2), or on a 100 kDa MWCO Biotech Cellulose Ester (CE) membrane for larger reactions (Spectrum Chemical Mfg Corp, New Brunswick, NJ, USA). Conjugated and unconjugated nanoparticles, larger than 100 kDa, were retained. 

### 2.5. SDS-PAGE for Quantification of the Coupling Efficacy and Western Blotting to Identify Components

After the coupling reactions, a sample was collected and supplemented with 5× SDS-PAGE loading buffer. Samples were denatured at 95 °C for 10 min; 0.1 µg samples were then loaded on a 15% Tris–glycine PAGE. Proteins on the gel were stained with Sypro-Ruby gel stain (ThermoFisher scientific, Waltham, MA, USA) following the manufacturer’s rapid staining instructions. Signal intensity was quantified using the image analysis software ImageJ (version 1.52a) (free software; https://imagej.net/Wayne_Rasband). The conjugation efficiency was defined as the quotient of the coupled PapMV CP signal over the sum of the coupled PapMV CP and the unconjugated PapMV CP signals. 

The identity of the components in each signal was revealed by Western blotting using an in-house anti-PapMV CP polyclonal antibody or an anti-Influenza A virus M2 protein antibody (Abcam, Toronto, ON, Canada) followed by incubation with anti-rabbit antibodies coupled with alkaline phosphatase (Jackson Immunoresearch, West Grove, PA, USA,) using step 1 nitro blue tetrazolium chloride (NBT)/5-bromo-4-chloro-3-indolyl phosphate (BCIP) as a revealing reagent (ThermoFisher scientific, Waltham, MA, USA).

### 2.6. Animals, Immunization, and Immune Response Quantification 

#### 2.6.1. Immunization and M2e Antibody Titration by ELISA

Six to 10 week old female BALB/c mice were immunized by intramuscular injection (i.m.) twice, 3 weeks apart. Each group contained five mice. Group 1 received control buffer (50 μL of 10 mM Tris–HCl, 150 mM NaCl pH 8.0); Group 2 received PapMV-C and free M2e peptide; Group 3 received PapMV-C coupled to M2e; Group 4 received PapMV-N and free M2e peptide; Group 5 received PapMV-N coupled to M2e. To assess the humoral response to the M2e peptide, blood samples were collected before each boost-immunization on days 20 and 42. Serum was separated from the blood by centrifugation in BD Microtainer SST blood collection tubes (BD, East Rutherford, NJ, USA) for 2 min at 10,000× *g*. The immunoglobulin G (IgG) 2a endpoint titer to M2e peptides in the sera of immunized mice was determined by enzyme linked immunosorbent assay (ELISA) as described previously [33]. Briefly, 96-well flat-bottom nunc^TM^ MaxiSorp plates (VWR, Radnor, PA, USA) were coated overnight at 4 °C with the M2e peptide at 1 μg/mL. Two-fold step serial dilutions of mice sera, starting at 1 in 50, were prepared. Primary antibodies were revealed using peroxidase-conjugated goat anti-mouse IgG2a (Jackson Immunoresearch, West Grove, PA, USA). Results are expressed as antibody endpoint titers greater than threefold OD_450nm_ of the background value consisting of a pool of pre-immune sera.

#### 2.6.2. Immunization and IFN Detection by ELISPOT Directed to the NP Peptide

Six to 10 week old female BALB/c mice were immunized by i.m. injection, twice, 3 weeks apart. Each group contained 5 mice. Group 1 received control buffer (100 μL of 10 mM Tris–HCl, 150 mM NaCl pH 8.0); Group 2 received PapMV-C combined with NP peptide; Group 3 received PapMV-C coupled to NP; Group 4 received PapMV-N combined with NP peptide; and Groups 5–7 received PapMV-N coupled to three increasing density of NP. Two weeks after the second immunization, mice were terminated and their spleens harvested to conduct an ELISPOT assay as previously reported [29]. Reactivation of splenocytes was performed using Influenza NP peptide ^147^TYQRTRALV^155^ (Anaspec, Fremont, CA, USA) on 500,000 cells. The precursor frequency of specific T cells was determined by subtracting the number of background spots from cells stimulated with media alone from the number of spots observed in wells containing cells reactivated with NP peptide. 

#### 2.6.3. Immunization and Interferon Alpha (IFN) Quantification

BALB/C mice, five per group, were immunized, once, via the intravenous route (i.v.) with 200 µg of PapMV or PapMV-N or PapMV-C nanoparticles. At 6 or 8 h after immunization, the animals were terminated, and their blood was harvested by heart puncture. The level of IFN in animal sera was assessed using the VeriKineTM Mouse IFN Alpha ELISA Kit using a sandwich immunoassay directed to IFN⍺, according to the manufacturer’s protocol (PBL Assay Science, Piscataway, NJ, USA). 

### 2.7. Statistics

Data were analyzed with Graph Pad PRISM 7.0d software (GraphPad Software, Inc., San Diego, CA, USA) for statistical significance using the ANOVA test. Tukey’s post hoc test was also used to compare the difference among groups of mice. Values of *p* smaller than 0.05 were considered significantly different.

### 2.8. Ethics Statement

All animal work was previously approved by the “Comité de Protection des Animaux—CHUQ(CPA-CHUQ)” of the institution. The authorization number was 19-003-1.

## 3. Results

### 3.1. Design, Production, and Characterization of PapMV Nanoparticles

The main objective of this study was to develop a novel PapMV vaccine platform where the coupling of the peptide with the SrtA was done at the N-terminus of the PapMV nanoparticle (PapMV-N). The immunogenicity was compared with an older version of the PapMV vaccine platform (PapMV-C) where the fusion was performed at the C-terminus. The fusion at the C-terminus allowed coupling of the peptides only at each extremities of the nanoparticle, limiting the levels of coupling at approximately 20% [19]. However, coupling at the N-terminus of the PapMV CP with PapMV-N was expected to reach higher levels of coupling since the N-terminus is freely available on all the surface of the PapMV nanoparticle [33].

Antigen coupling to the PapMV nanoparticle was performed using sortase A (SrtA)—a bacterial transpeptidase that generates a covalent link between two proteins through recognition of the LPETGG donor and G-repeat acceptor motifs [32]. Engineering of PapMV CP to produce a PapMV-C nanoparticle harboring the LPETGG SrtA donor motif at the C-terminus was previously reported [33,35]. Coupling of the antigen with SrtA was performed using peptide antigens that present the G-repeat motif to their N-terminus. To produce the PapMV-N nanoparticle, the G-repeat donor motif was inserted at the N-terminus of the CP. In this case, the nanoparticle was coupled via SrtA using peptide antigens presenting the LPETGG receptor motif at their C-terminus. More specifically, the PapMV-N sequence was fused genetically to the C-terminus of the intein to replace the CP N-terminal methionine with four glycines (GGGG), leading to a fusion protein (Figure 1A). The fusion protein (49.3 kDa) was produced in *E. coli* and purified on IMAC (Figure 1B). The intein’s proteolytic activity for self-cleavage of the fusion protein was activated by decreasing the pH from 8.0 to 6.5 with incubation at 22 °C. This reaction generated two distinct products: PapMV-N CP (24.1 kDa) and intein (25.2 kDa) (Figure 1B). The PapMV-N CP was then further purified on IMAC, leading to a highly purified PapMV-N CP (>95%). PapMV-N CP was used to generate the PapMV-N nanoparticles through self-assembly onto a noncoding ssRNA template. 

The shape of the PapMV-N nanoparticle was undistinguishable from nanoparticles assembled using the PapMV CP or the PapMV-C CP as observed by electron microscopy (EM) (Figure 2A). The mean length of the three types of nanoparticles was approximately 80 nm (Figure 2B). The heat denaturation profile revealed that both PapMV and PapMV-N nanoparticles started aggregating at approximately 40 °C, while the PapMV-C nanoparticle remained stable up to 50 °C (Figure 2C). The increasing length of the particles at high temperature was probably induced by partial denaturation of the CP, leading to nonspecific aggregation. 

### 3.2. Assessment of the Humoral Response Induced by the Vaccine Platforms Coupled to the M2e Peptide

To assess the humoral response induced by the PapMV-C and PapMV-N vaccine platforms, we used the influenza M2e peptide as the reference antigen. The M2e peptide is derived from the extracellular domain of matrix protein 2 (M2e) of influenza virus. M2e is highly conserved in most influenza A strains [37] and is a valuable antigen for inducing protection to influenza infection [28]. The M2e peptide coupled to the PapMV-C vaccine platform anchors the SrtA acceptor motif (G-repeat) to its N-terminus, while the M2e coupled to the PapMV-N vaccine platform anchors the LPETGG donor motif at its C-terminus. The maximum coupling efficiency of the M2e peptide on PapMV-C was 19% (PapMV-C/M2e 19%) (Figure 3A, top panel). To compare the efficacy of the two vaccine platforms, a similar level of coupling (14%) was obtained on the PapMV-N vaccine platform (PapMV-N/M2e 14%) (Figure 3A, top panel). SDS-PAGE analysis of the coupling reactions confirmed that a signal with a higher molecular weight than the PapMV-C or PapMV-N CP was observed (Figure 3A, top panel). The signal also reacted with the specific M2e and PapMV CP antibodies by Western blotting, thereby confirming the identity of the coupled fusion protein (Figure 3A, middle and lower panels). 

Immunization of mice was performed with the formulation buffer alone (10 mM Tris–HCl, 150 mM NaCl pH 8.0) and 10 µg of each platform coupled to the M2e peptide. The amount of M2e peptide coupled to 10 µg of PapMV-C/M2e 19% was estimated to be approximately 0.2 µg of M2e peptide. We added 1 µg of free M2e peptide in the uncoupled formulations with the PapMV-C or the PapMV-N platforms, corresponding to a fivefold excess of the M2e peptide as compared to the groups coupled to the M2e peptide. ELISA analysis of serum harvested after a single immunization revealed that only the formulations where the peptide is coupled to the nanoparticle induced a strong IgG2a humoral response to M2e (Figure 3B); this observation suggests that the peptide has to be coupled to the vaccine platform to induce production of antibodies. Interestingly, the PapMV-N/M2e 14% vaccine induced IgG2a titers 16-fold higher than those induced by PapMV-C/M2e 19%. ELISA analysis of serum after boost immunizations confirmed the higher efficacy of the PapMV-N/M2e 14% vaccine formulation (Appendix A).

### 3.3. Assessment of Humoral Response to M2e Peptide with Nanoparticles Coupled at Increasing Density

The density of peptide coupling to the PapMV-N nanoparticles was increased from 14% to 28%, 44%, or 83% (Figure 4A) by increasing the coupling reaction period and increasing the concentration of M2e peptide. This was not possible with the PapMV-C nanoparticle, which reached maximal coupling at approximately 20% (Figure 3A). Coupling efficiency was assessed on SDS-PAGE by quantifying signal intensity relative to the 25 kDa MW marker (Figure 4A, top panel, lanes 1–4). Positive signals were seen with specific M2e and PapMV CP antibodies on Western blot, thus confirming the identity of the coupled fusion protein (Figure 4A, middle and lower panels). 

Mice were immunized once with 10 µg of the PapMV-N platform coupled with increasing amounts (14%, 28%, 44%, and 83%) of M2e peptide. Groups receiving the vaccine formulation buffer alone (buffer) or the PapMV-N platform combined with free M2e peptide (uncoupled) were used as controls. The amount of M2e peptide coupled to 10 µg of PapMV-N/M2e at the 83% concentration was estimated at approximately 0.9 µg. Therefore, in the uncoupled control, we added 1 µg of free M2e peptide together with the PapMV-N platform. ELISA analysis of serum harvested 3 weeks after a single immunization revealed that the IgG2a humoral response to M2e increased with the density of M2e peptide coupled to the nanoparticles (Figure 4B). The antibody titers were 8.6, 9.2, 11, and 11.4 for coupled PapMV-N at 14%, 28%, 44%, and 83%, respectively. The antibody titers of the two groups with the highest density of peptides (PapMV-N/M2e 44% and 83%) were significantly higher than the titers obtained in the two groups with lower peptide density (PapMV-N/M2e 14% and 28%). ELISA analysis of serum harvested 3 weeks after a boost immunization led to high geometric mean titers in all groups where the peptide was coupled to the nanoparticles, suggesting that the immune response reached a saturation limit (Appendix A).

In the experimental work described in Figure 4B, the amount of PapMV-N (10 µg) administered to the mice was similar in each group, while the amount of the M2e peptide coupled to the nanoparticles increased between groups. Therefore, the PapMV-N/M2e 83% was decorated with 5.9 times more M2e peptide than the PapMV-N/M2e 14%, and with three times more than the PapMV-N/M2e 28%. To assess whether the density of coupled peptide has an impact on the immunomodulation potency of the PapMV platform, mice were immunized with 10 µg of PapMV-N/M2e 14%, 5 µg of PapMV-N/M2e-28%, and 1.7 µg of PapMV-N/M2e 83%; the humoral response was analyzed by ELISA (Figure 4C). At 20 days following immunization, IgG2a titers to M2e peptide were similar in all groups, suggesting that the density of peptide coupled to the nanoparticles does not impact the immunomodulation potency of the antigen-presenting platform. After boost immunization, the antibody titers to M2e were similar in all groups (Appendix A).

### 3.4. Assessment of CD8+-Mediated Response Induced by the Vaccine Platforms and Impact of Coupling Density on Immunogenicity

Our next objective was to assess the efficacy of PapMV-C and PapMV-N vaccine platforms to elicit a CD8+-mediated immune response to a CTL epitope. The impact of coupling density on CTL response was also assessed. The CTL epitope TYQRTRALV flanked by five native amino acids derived from the nucleocapsid (NP) of the influenza A virus at its N- and C-terminus was chosen as our reference epitope [30]. The NP peptide was coupled to the two different vaccine platforms. The maximum coupling density obtained on the PapMV-C platform was 30% (Figure 5A). Coupling at 15%, 30%, and 60% was obtained on the PapMV-N platform (Figure 5A). Animals were immunized with the different constructs with vaccine formulation buffer serving as a negative control. ELISPOT assay revealed the numbers of CD8+ T cells secreting IFN Figure 5B). Interestingly, only the formulations coupled to the PapMV-N at 30% and 60% led to detection of a significant amount of CD8+-specific T-cells secreting IFN (Figure 5B). This experiment confirmed that the PapMV-N vaccine platform is more effective than the PapMV-C platform at triggering a CD8+-mediated immune response. It also confirmed that a higher density of coupling (30% and 60%) led to a more robust CD8+-mediated T-cell response than coupling at lower density (15%).

### 3.5. The PapMV-N Platform Is a Stronger Inducer of Type I Interferon than PapMV-C

The PapMV-N platform was shown superior to PapMV-C in the induction of both the humoral (Figure 3B) and the CTL (Figure 5B) immune response directed to the coupled peptide. This suggests that there is an enhancing factor associated with the PapMV-N platform that appears to boost the immune response. PapMV nanoparticles were previously shown to be a strong inducer of secretion of type I interferon (IFN acting though the stimulation of TLR 7 [7,8,9] and 8 [27]. In an attempt to evaluate whether the difference in immunogenicity between the two vaccine platforms is linked to the capacity to induce secretion of IFN, we immunized Balb/C mice once (i.v.) with 200 µg of PapMV, PapMV-N, or PapMV-C, and we assessed the levels of IFN secreted in the blood at 6 h and 8 h after treatment (Figure 6). As expected, PapMV triggered high levels of IFN at 6 h and 8 h after injection. The PapMV-N nanoparticle was significantly less efficient than the PapMV nanoparticle (no Srt receptor motif on CP) but was significantly more efficient than PapMV-C, which failed to induce IFN.

## 4. Discussion

We compared two different vaccine platforms based on the same backbone: PapMV-C and PapMV-N. The two vaccine platforms showed comparable biophysical properties and shared the same length and appearance as observed with DLS and EM (Figure 2A,B). Interestingly, PapMV-C was the only nanoparticle that showed stability at temperatures exceeding 40 °C and was also the only nanoparticle that failed to induce secretion of IFN after i.v. injection (Figure 6). PapMV was previously shown to induce secretion of IFN through the stimulation of TLR7 in plasmacytoid dendritic cells (pDCs) when injected i.v. [25]. In our model of activation of TLR7, nanoparticles became internalized in pDCs, reaching the endosome compartment [27], where the harsh conditions induced disassembly of the nanoparticle, leading to the release of the ssRNA, activation of TLR7 and, consequently, IFN secretion (Figure 7). It is likely that PapMV-C failed to induce secretion of IFN because of its greater stability, which enabled the nanoparticle to resist the harsh conditions of the endosome, avoiding the rapid release of the ssRNA and the activation of TLR7. Therefore, PapMV-C is a weaker immune enhancer and, consequently, a less efficient vaccine platform than PapMV-N. 

According to the three-dimensional (3D) model of PapMV CP [33], the LPETGG motif of the PapMV-C vaccine platform is localized inside the nanoparticles and exposed at the surface only at the extremities of the nanoparticles. However, the N-terminus of the CP is predicted to be available over the entire surface of the nanoparticle [33]. Our data confirm this prediction, since a threefold higher level of coupling was obtained with the PapMV-N vaccine platform. Our data reveal that high coupling leads to a stronger humoral and CTL immune response against the coupled peptide, giving the PapMV-N platform a second advantage over the PapMV-C platform. This feature allows the design of vaccines using fewer nanoparticles to obtain the same efficacy, making this approach more economic. 

The PapMV-N vaccine platform is a versatile tool that allows the rapid development of vaccine candidates. Such tools can be especially valuable in the context of a pandemic, allowing several peptide antigens to be screened rapidly. Because of their remarkable stability (>7 years in solution at 4 °C (Appendix A), stockpiling of large quantities of PapMV nanoparticles is easily accomplished. Similarly, PapMV-N was demonstrated to be stable for more than 8 months, and this nanoparticle is expected to be as stable as PapMV (Appendix A). In preparation for new viral threats, it is conceivable to stockpile a large number of nanoparticles ready to be coupled with a relevant peptide antigen. The attachment of peptides to the nanoparticle surface using SrtA can be done rapidly with peptides designed from the genetic sequence of the pathogen, producing several candidate vaccines ready to be tested in animal models for their efficacy to induce protection against the viral threat. Given that the coupling reaction with SrtA is flexible and efficient with almost any soluble peptide, including large peptides, such as the M2e peptide (29 amino acids), new vaccine candidates can be developed rapidly. In addition, the selection of highly conserved regions from the emerging virus, such as the viral nucleocapsid, can potentially lead to vaccine candidates capable of providing broad protection that can slow down the spread of the virus or mutated variants. 

## 5. Conclusions

In brief, we have showed that the SrtA mediated attachment of peptide antigens at the N-terminus of the PapMV CP allowed to reach higher levels of coupling on the nanoparticle than when it is performed at the C-terminus. Consequently, PapMV nanoparticles harboring high peptide density at their surface are more effective to elicit the humoral and the cellular mediated immune response to the coupled peptide antigens. The PapMV-N adjuvant property was also showed to be superior to the PapMV-C vaccine platform based on its capacity to induce secretion of IFN through the stimulation of the TLR7/8.

Considering the current global pandemic and the need for rapid, adaptable, vaccine development into sharp focus, we believe that our novel nanoparticle vaccine platform can make a valuable contribution to this field.

## Figures and Tables

**Figure 1 vaccines-09-00033-f001:**
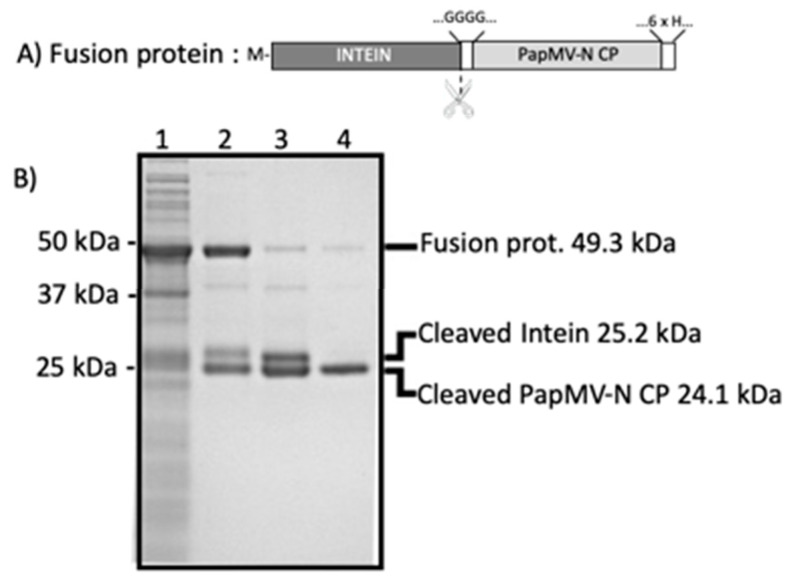
Purification of papaya mosaic virus with peptide at the N-terminus (PapMV-N) coat protein (CP). (**A**) Schematic representation of the intein–PapMV-N CP construct cloned in the pTWIN1 vector and expressed in *Escherichia coli*. The N-terminal methionine of the fusion protein is shown with a M. The site of cleavage by the intein is illustrated with scissors. Upon cleavage, the N-terminus of the PapMV-N CP presents GGGG at its N-terminus. The PapMV-N CP harbors a 6His-tag at its C-terminus to ease the purification process. (**B**) SDS-PAGE revealing the protein content of samples taken at successive purification steps of the PapMV-N CP. Molecular weight markers are shown to the left. Signals corresponding to the fusion protein (49.3 kDa), the cleaved intein (25.2 kDa), and the cleaved PapMV-N CP (24.1 kDa) are indicated on the right. Lanes: *1* lysate of induced bacteria producing the recombinant fusion protein; *2* fusion protein purified on ion matrix affinity chromatography (IMAC); *3* cleavage products; *4* purified PapMV-N CP.

**Figure 2 vaccines-09-00033-f002:**
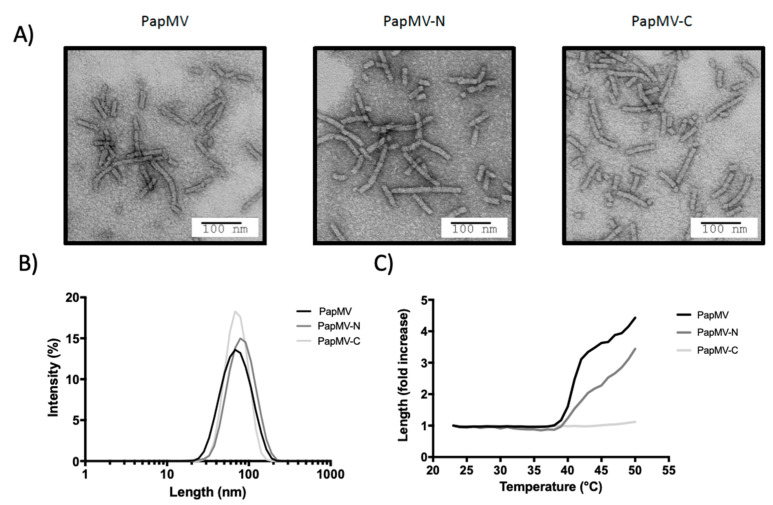
Characterization of PapMV, PapMV-C, and PapMV-N nanoparticles. (**A**) Electron micrographs of nanoparticles. (**B**) Dynamic light scattering (DLS) shows that the three types of nanoparticle have a similar average length of 70–80 nm. (**C**) The DLS was performed at increasing temperature to assess the resistance to heat denaturation of each nanoparticle.

**Figure 3 vaccines-09-00033-f003:**
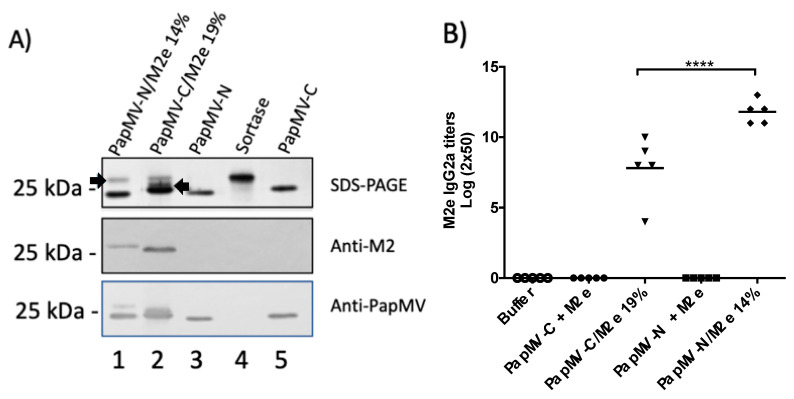
Coupling of the M2e peptide to the PapMV vaccine platforms and assessment of the humoral response directed to M2e in mice. (**A**) The M2e antigen was coupled to the PapMV-N and PapMV-C vaccine platforms using the sortase A (SrtA) enzyme. Coupled bands for both platforms can be visualized on SDS-PAGE slightly over the 25 kDa marker. The arrow pointing to the right indicates the signal of the PapMV-N coupled to the M2e peptide (lane 1), and the arrow pointing to the left indicates the signal of the PapMV-C coupled to the M2e peptide (lane 2). Signals corresponding to PapMV-N CP (lane 3) and PapMV-C CP (lane 5) are also observed. The signal corresponding to SrtA is shown in lane 4. Remaining SrtA after the coupling reaction is seen in lanes 1 and 2. Proteins from the SDS-PAGE were transferred to a membrane to perform Western blotting using anti-M2e (middle panel) or anti-PapMV (lower panel) antibodies. (**B**) Balb/C mice, five per group, were immunized once, intramuscularly (i.m.), with formulation buffer (Buffer), PapMV-C (10 µg) with 1 µg of free peptide M2e, 10 µg PapMV-C/M2e 19%, PapMV-N (10 µg) with 1 µg of free peptide M2e, and 10 µg PapMV-C/M2e 14%. ELISA was performed with serum harvested at day 20 to assess immunoglobulin G (IgG) 2a titers directed to the M2e peptide. **** *p* > 0.0001.

**Figure 4 vaccines-09-00033-f004:**
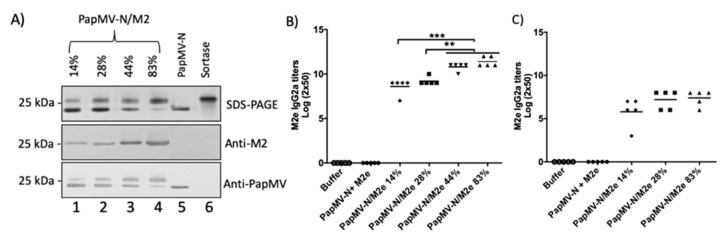
Impact of coupling density on the humoral response of the M2e peptide on PapMV-N nanoparticles. (**A**) Various coupling densities (14%, 28%, 44%, and 83%) of the M2e peptide on the PapMV-N vaccine platform were generated. Coupling was visualized on SDS-PAGE (top panel). Lanes 5 and 6 correspond to uncoupled platform and sortase A, respectively. The proteins were transferred to a membrane for Western blotting with either anti-M2e (middle panel) or anti-PapMV (lower panel) antibodies. (**B**) Humoral response obtained using the same amount of platform with increasing amount of coupled M2e peptide. Balb/C mice, five per group, were immunized once via the intramuscular route (i.m.) with vaccine formulation buffer (Buffer), PapMV-N (10 µg) with 1 µg of free peptide M2e, or 10 µg of PapMV-N platform coupled to the M2e peptide with an efficiency of 14%, 28%, 44%, or 83%. ELISA analysis of serum harvested at day 20 was used to assess IgG2a titers directed to the M2e peptide. (**C**) Humoral response obtained when immunization was performed with the same amount of M2e peptide but with decreasing amounts of platform. Balb/C mice, five per group, were immunized once i.m. with the same amount of M2e peptide presented at different densities on the PapMV-N platform. The different groups of mice received formulation buffer as a negative control (Buffer), 10 µg of PapMV-N platform coupled with an efficiency of 14% (PapMV-C/M2e 14%), 5 µg of PapMV-N platform coupled with an efficiency of 28% (PapMV-N/M2e 28%), and 1.7 µg of PapMV-N platform coupled with an efficiency of 83% (PapMV-MN/M2e 83%). ELISA analysis of serum harvested at day 20 was used to assess IgG2a titers directed to the M2e peptide. ** *p* > 0.01, *** *p* < 0.001.

**Figure 5 vaccines-09-00033-f005:**
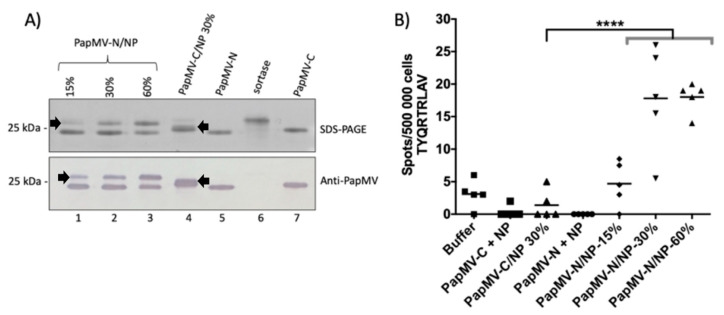
Impact of coupling density of a CTL epitope on the level of CD8+-mediated immune response. (**A**) Visualization of coupling of the nucleocapsid (NP) CTL epitope derived from the nucleocapsid of influenza A virus to PapMV-C and PapMV-N vaccine platforms on SDS-PAGE (top panel). Coupling efficiencies of 15% (lane 1), 30% (lane 2), and 60% (lane 3) on the PapMV-N platform are shown. Coupling of the NP peptide on the PapMV-C platform at 30% is shown in lane 4. The arrowheads pointing to the right indicate the position of PapMV-N coupled to NP peptide, and the black arrowheads pointing to the left indicate the position of PapMV-C coupled to the NP peptide (lane 4). The PapMV-C CP (lane 5), sortase (lane 6), and the PapMV-C CP (lane 7) are also shown. The proteins from SDS-PAGE were transferred to a membrane for Western blotting with anti-PapMV antibodies (lower panel) to confirm the identity of the coupled and uncoupled bands. (**B**) CD8+-mediated immune response directed to the NP CTL epitope coupled on the PapMV vaccine platforms. The influenza CTL epitope (TYQRTRLAV), named NP, was fused to PapMV-C and PapMV-N and injected i.m. in Balb/C mice, five per group, at days 0 and 21. The animals were injected with vaccine formulation buffer, PapMV-C (30 µg) + NP (1 µg), 30 µg of PapMV-C/NP 30%, PapMV-N (30 µg) + NP (2 µg), 30 µg of PapMV-N/NP 15%, 30 µg of PapMV-N/NP 30%, or 30 µg of PapMV-N/NP 60%. At day 35, spleens were harvested and the CD8+-mediated immune response was assessed by ELISPOT using the TYQRTRLAV peptide to stimulate the splenocytes. **** *p <* 0.0001.

**Figure 6 vaccines-09-00033-f006:**
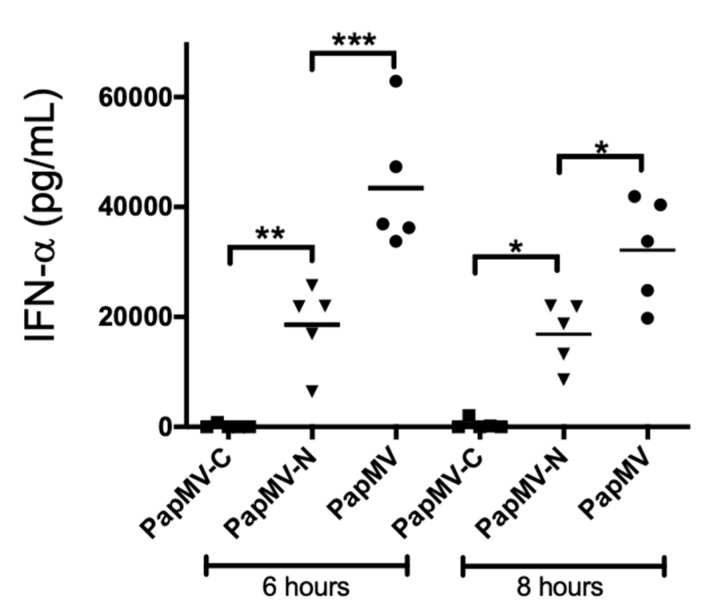
The PapMV-N vaccine platform is a strong inducer of interferon (IFN) secretion. Balb/C mice, five per group, were immunized i.v. with 200 µg of PapMV, PapMV-N, or PapMV-C. Blood was harvested at either 6 h or 8 h after injection, and ELISA directed to serum IFN was performed. * *p* < 0.05, ** *p* < 0.01, and *** *p* < 0.001.

**Figure 7 vaccines-09-00033-f007:**
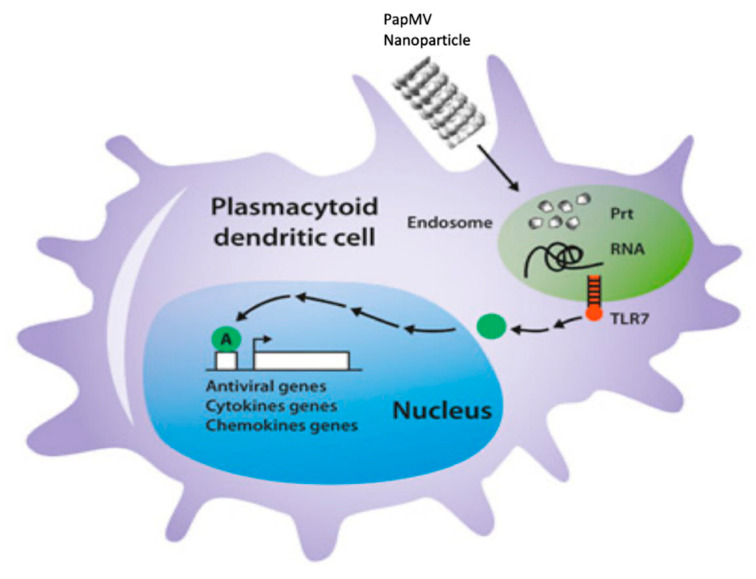
Schematic representation of the mechanism of activation of Toll-like receptor 7 (TLT7) by a PapMV nanoparticle. In brief, PapMV nanoparticles are internalized into immune phagocytes, such as plasmacytoid dendritic cells (pDCs) or macrophages, and reach the endosomal compartment, where they become disassembled because of the harsh conditions in the endosome. The single-stranded RNA (ssRNA) contained in the nanoparticle is liberated and activates TLR7, triggering an antiviral immune response.

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
