# Peer review of "Modulation of Antigen Display on PapMV Nanoparticles Influences Its Immunogenicity"

_vaccines, 2021, doi:10.3390/vaccines9010033_

Round 1

Reviewer 1 Report

Laliberte-Gagne and colleague examined the papaya mosaic virus nanoparticle as a vaccine platform.  Dr. Leclerc, the corresponding author, has published a number of papers in this area, and this paper is a logical continuation of previous projects from his lab.  The current manuscript is examines particle stability and the effects on antigenicity of attaching the vaccine antigen to the amino terminus vs the carboxy terminus of the PapMV coat protein via sortase-mediated coupling and examining coupling density.  Fig6  in conjunction with Fig2 is exceptionally interesting.  These two figures by themselves will generate a lot of questions and provide much material for future work.  The figures are well laid out and include all the necessary controls. 

The paper is essentially ready for publication.  I’ve made some minor comments below that can be addressed at the authors’ discretion.

Minor Comments:

  1. Fig2 examines the effect of conjugation location (N-term vs C-term) on the antibody response to M2e. In this case, Conjugation to the N results in a better antibody response.  Analysis of the antibody response to a second antigen would strengthen the paper.  As the manuscript is currently, we don’t know how well this result generalizes to other antigens.  Granted one additional antigen wouldn’t completely resolve this weakness, but it would provide some additional information. 
  2. Lines 78 & 84 – do not capitalize “Intein”
  3. Fig 1A – Cartoon appears to show the “M” terminus of the protein. I think this should be “N”.  Also, I think you should add a “-C” to the left end of the figure to make it perfectly clear.  I realize now that “M” is for methionine, so this change is at the authors’ discretion, though I think marking the amino and carboxy termini would make the figure work better with the text.
  4. Fig 2B – X-axis labels are unclear.
  5. Fig 4 legend should be indented to be consistent with other legends.

Author Response

Reviewer 1

  • Fig2 examines the effect of conjugation location (N-term vs C-term) on the antibody response to M2e. In this case, Conjugation to the N results in a better antibody response.  Analysis of the antibody response to a second antigen would strengthen the paper.  As the manuscript is currently, we don’t know how well this result generalizes to other antigens.  Granted one additional antigen wouldn’t completely resolve this weakness, but it would provide some additional information. 

Dear reviewer, thanks for your comment.

First of all, based on your comments, I believe you are referring to Fig. 3 and not Fig. 2.

I understand your point and this is why we tested two different antigens, the M2e peptide and the NP peptide and compared their immunogenicity on the two different platforms. The M2e peptide is a B cell epitope while the NP peptide is a CTL epitope. Considering that both peptides, that are triggering two different type of immune response (humoral vs CTL), were more immunogenic when attached to the PapMV-N platform, we concluded that we have an argument to suggest that the PapMV-N platform is more efficient than the PapMV-C platform. In addition this is also supported by Fig.6 that demonstrates the higher efficiency of the PapMV-N platform to elicit the IFN-a secretion, suggesting that PapMV-N is a much better adjuvant than PapMV-C.

In addition, it is possible that we might see some variations in the humoral immune response observed against different coupled peptides depending of their nature, length, amino acid composition, etc….I totally agree with you on this point but this is a very complex question that cannot be answered with the data presented in this manuscript. To really answer your question, many years of experiments with more than 30 different peptides will be needed. This will help us to draw a pattern that will help us to predict which kind of peptide is more effective to mount an humoral response when coupled on the PapMV-N vaccine platform.

            To illustrate this concept, the Fig. 1 shows the humoral response directed to three different peptides derived from the SARS-CoV-2. As you can see, the strength of the humoral response to the three peptides differs and depends on the nature of the peptide. Interestingly, there is more variations when animals are vaccinated with the PapMV-N/S1 peptides. We do not understand for now the reasons behind this variation but this will be investigated by our group in the next few years. Intra-group variations are less important for the M and the S2 peptides. The results also reveal that the PapMV-N platform is versatile and efficient with different type of peptides. We do not wish to include these results in the manuscript since we are currently preparing another manuscript on a candidate vaccine against SARS-CoV-2.

Figure 1. (see attached file)

  • Lines 78 & 84 – do not capitalize “Intein”

Thanks, we fixed it

  • Fig 1A – Cartoon appears to show the “M” terminus of the protein. I think this should be “N”.  Also, I think you should add a “-C” to the left end of the figure to make it perfectly clear.  I realize now that “M” is for methionine, so this change is at the authors’ discretion, though I think marking the amino and carboxy termini would make the figure work better with the text.

Dear reviewer, thank you for your comment, but we prefer to keep the figure in its current form. It is implicit that the right end of the protein is the N-terminus and the left end, the C-terminus. It is not needed to add it to the figure to our point of view. We added on the legend to the figure that the M correspond to the first amino acid at the N-terminus of the protein for clarity.

  • Fig 2B – X-axis labels are unclear.

Dear reviewer, thank you, we have fixed it.

  • Fig 4 legend should be indented to be consistent with other legends. 

Dear reviewer, we change the title of Fig 4 to make it consistent with the title of the Fig. 5. We also change for the format palatino 9, like for the other legends. Thanks.

Reviewer 2 Report

The author reported the effects of the peptide antigen at the surface or at the extremities of the nanoparticles on the immune response.The author synthsis and characterized the PapMV nanoparticles. they assessed the humoral response induced by the vaccine platforms coupled to the M2e pettide and the petptide density, et al. it is intresesting work and can be published in Vaccines.

Author Response

No revision needed, reviewer @ accept the manuscript

Thanks

Reviewer 3 Report

The ms revisits the papaya mosaic virus (PapMV) vaccine platform in which PapMV particles formed with the viral coat proteins fused to a peptide antigen result in a stronger immune response. Previous work from the same group established the efficacy of antigen fusion to the C-terminal end of the PapMV coat proteins. In this paper, the authors repeated the study this time with a N-terminal end fusion.  The paper is straightforward. Some additional experimental details should be added.

Ln 67. What were the limitations of the C-terminal fusion to justify the exploration of the N-terminal end of the coat protein?

Ln 216:  what is the biological relevance of the aggregation? Is it important for the PapMV particles role in activation of the humoral response?

Ln 232-234: what were the conditions to these maximum coupling efficiencies of 19% and 14% respectively? Was it based on previous formation from the first study for the C-terminal fusion?

In the same line of thought, ln 268:  what were the changes in reaction period and concentration of M2e.  include those details also Fig 4A legend for lane 1-2-3-4

Ln 239, were these coupled PapMV-N/C-M2e purified (to remove the uncoupled peptides) prior to injection into the mice? Or was it the entire formulation that was injected into the mice?

Author Response

Reviewer 3

Ln 67. What were the limitations of the C-terminal fusion to justify the exploration of the N-terminal end of the coat protein?

Dear reviewer, thank you for your comment. For clarification, we added the paragraph below at the section 3.1 to ease the understanding of the readers.

‘The main objective of this study is to develop a novel PapMV vaccine platform where the coupling of the peptide with the SrtA will be done at the N-terminus of the PapMV nanoparticle (PapMV-N). The immunogenicity will be compared with an older version of the PapMV vaccine platform (PapMV-C) where the fusion is performed at the C-terminus. The fusion at the C-terminus allows coupling of the peptides only at each extremities of the nanoparticle, which is limiting the levels of coupling at approximately 20% [19]. However, coupling at the N-terminus of the PapMV CP with PapMV-N is expected to reach higher levels of coupling since the N-terminus is freely available on all the surface of the PapMV nanoparticle [19]. ‘

Ln 216:  what is the biological relevance of the aggregation? Is it important for the PapMV particles role in activation of the humoral response?

Dear reviewer, the aggregation of the nanoparticle induced by heat is a read-out that reveal the denaturation of the PapMV CP. The denatured protein exposes hydrophobic residues to the solvent that lead to aggregation of the proteins. It is for us an efficient way to assess nanoparticle stability. As mentioned in the discussion, end of the first paragraph: ‘It is likely that PapMV-C failed to induce secretion of IFNa because of its greater stability, which enables the nanoparticle to resist the harsh conditions of the endosome, avoiding the rapid release of the ssRNA and the activation of TLR7. Therefore, PapMV-C is a weaker immune enhancer and consequently a less efficient vaccine platform than PapMV-N’.

Therefore, to be a good adjuvant, we hypothesized that the nanoparticle need to be stable upon injection, but also need to become unstable and easy to open up when it is internalized into the endosome of the APCs to liberate the ssRNA and activate the TLR7/8. The PapMV and PapMV-N appear to satisfy both of these conditions. The stability of PapMV-C at high temperature  is probably linked to a higher stability in the endosome leading to a less efficient stimulation of TLR7/8 since the ssRNA cannot be released from the nanoparticle. Of course, this is speculative but it is our lead hypothesis to explain the difference between the adjuvant properties of the PapMV-N and the PapMV-C.

Ln 232-234: what were the conditions to these maximum coupling efficiencies of 19% and 14% respectively? Was it based on previous formation from the first study for the C-terminal fusion?

Dear reviewer, the maximum coupling efficiency of the M2e peptide on the PapMV-C vaccine platform is 19%. Coupling conditions were similar to those used in our previous study (Therien et al., 2017).

            In an attempt to get a similar coupling efficiency on PapMV-N,  we have modified the conditions of the sortagging reaction and obtained 14% of coupling efficiency on the PapMV-N. The maximum of coupling that could be obtained on the PapMV-N was 83% as showed on Fig. 4.

In the same line of thought, ln 268:  what were the changes in reaction period and concentration of M2e.  include those details also Fig 4A legend for lane 1-2-3-4

Dear reviewer, we have omitted to include the details of the modifications made in the sortagging reactions in the text to focus on the immune response and keep the text simple.  In brief, the level of coupling on the PapMV-C were always the maximum of coupling that we could obtained. However, to get different levels of coupling on the PapMV-N platform, we have modified the ratio in µM/µM between the PapMV-N nanoparticles and the peptide, the amount of SrtA added to the reaction and the time of incubation. See the table below that summarize the different conditions used for the sortagging with the SrtA:

Formulation 

Amount peptides (µM) 

Amount Sortase (µM) 

Amount platform (µM) 

Time of reaction (h) 

PapMV-N/14%

25 

25 

23 

PapMV-N/28%

37.5 

50 

25 

2.5 

PapMV-N/44%

37.5 

25 

25 

23 

PapMV-N/83%

250 

25 

25 

23 

PapMV-C/19%

50 

50 

25 

2.5 

Ln 239, were these coupled PapMV-N/C-M2e purified (to remove the uncoupled peptides) prior to injection into the mice? Or was it the entire formulation that was injected into the mice?

Dear reviewer, all the sortagging reactions were dialyzed against the buffer Tris-HCl pH8.0 + 150mM NaCl with a membrane cutoff of 100kDa to remove the SrtA (approximately 25kDa) and the free peptides. The negative control were made of the same sortagging reactions without the peptide that were dialyzed as before and in which the peptide was added after the dialysis reaction + 1mM EDTA to inhibit putative sortagging reaction that might occur with small remaining amount of SrtA in the sample.